# A Hybrid Filtering Stage-Based Rotor Position Estimation Method of PMSM with Adaptive Parameter

**DOI:** 10.3390/s21144667

**Published:** 2021-07-07

**Authors:** Linxin Yu, Dazhi Wang

**Affiliations:** School of Information Science & Engineering, Northeastern University, Shenyang 110819, China; 1510309@stu.neu.edu.cn

**Keywords:** sensorless control, PMSM, phase-locked loop, rotor position estimation

## Abstract

The performance of sensorless control in a permanent magnet synchronous machine (PMSM) highly depends on the accuracy of rotor position estimation. Owing to its strong robustness, phase-locked loop (PLL) is widely used in rotor position estimation. However, due to the influence of harmonics existing in back electromotive force (EMF), estimation error occurs by using PLL. In this paper, a hybrid filtering stage-based PLL is proposed to improve the rotor position estimation. Adaptive notch filters and moving average filters are integrated together to eliminate harmonic EMF. To make the method effective under varying speed conditions, adaptive parameters design guidelines are provided, considering dynamic performance under a wide operating range. The proposed method can accurately detect rotor position even under harmonic EMF disturbances. It can also adjust the frequency adaptively based on the rotating speed of the rotor, which means the estimation performance is not deteriorated under rotating speed changing conditions. The simulation results verify the effectiveness of the proposed method.

## 1. Introduction

Due to the advantages of efficiency and high power density, permanent magnet synchronous motors (PMSM) have been widely used in electric vehicles, petroleum, and other industrial fields [1,2,3]. The control of PMSM relies on the rotor position measured by mechanical sensor of the motor. However, mechanical position sensors not only have the problems of installation difficulties and low reliability, but there is also a need to consider cost and installation space in practical applications. Therefore, sensorless control technology has become a hot spot in the research field [4,5].

Existing sensorless control technology can be classified into two categories: high-frequency signal injection method [6,7] and back-EMF estimation method [8,9]. The high-frequency signal injection method uses the salient pole effect of permanent magnet motor, which analyzes the position of the rotor by observing the current response under high-frequency voltage pulses. However, this method is only effective in low-speed operating conditions. The method of obtaining motor speed and position information by establishing the back-EMF is currently the most widely used method, which includes disturbance observing method, model reference adaptive method, and synovial observer method. Among them, the sliding mode observer method (SMO) based on back-EMF has the advantages of being less affected by changes in motor parameters and strong robustness, so it has been widely studied and applied [10,11].

Theoretically, SMO can only estimate the rotor position through discontinuous control and infinite frequency switching. However, due to the limitations of switching frequency and sampling frequency in reality, SMO unavoidably brings chattering problems to the control system. In order to improve the estimation accuracy of SMO, scholars have conducted a lot of research work. In [12], a sensorless vector control method based on SMO is proposed, which can track the rotor position and speed more accurately. It has better adaptive ability and robustness to speed and load changes. However, the traditional SMO used in this method still has the problem of chattering, in which observation accuracy is easily affected by the change of the motor speed. In [13], they theoretically analyzed the nonlinear voltage error caused by the capacitance of the switching device. The authors also point out that the current fluctuations during turn-on and turn-off will seriously affect the voltage error and cause fluctuation. The proposed method in the paper compensates the voltage errors by estimating the ripple currents. In [14], a phase estimation error compensation method based on current loop output error is proposed. The basic principle lies in the closed-loop adjustment of the compensation angle. Through the compensation of the error angle, the influence of the error on the DC voltage is reduced. This method avoids the secondary interference of voltage on position estimation, which ensures that the direct-axis voltage equation is always in an ideal balanced state, then realizes compensation of estimation errors. However, the above papers have ignored the harmonics in the electromotive force, which also may reduce the detection accuracy, especially for the low-order harmonics generated in unbalanced motor structure or unbalanced three-phase power supply [15]. If the influence of low-order harmonics is ignored, as the running time increases, the accumulation of this error will cause a larger deviation in the estimated rotor position.

Phase-locked loop (PLL) has become the most widely used phase detection method due to its accurate phase tracking capability and robustness [16,17]. Because of its excellent performance, it is also used in sensorless control. In [18], a fractional-order PLL (FO-PLL) technology is proposed to estimate rotor speed and position. FO-PLL provides additional degrees of freedom by selecting fractional coefficients and improves the dynamic performance; however, this method is only effective in the initial stage of motor control and under transient conditions. In [19,20], an improved PLL with low-cost switching Hall effect is proposed to estimate the speed and position of the rotor. Through the strategy, the rotor’s motion position error is kept within a small range. Although this method is claimed as a sensorless control strategy, it adopts a Hall-effect sensor, and still needs to consider the installation and the space occupied by the sensor. In [21], an angle and speed observer are designed. The observer uses the rotor flux vector to obtain the rotor position. After receiving the information from the observer, the rotor position is estimated through a PLL. In [22], an improved PLL named Q-PLL scheme based on quadrature components was proposed. It solves the speed reversal problem in the control of sensorless permanent magnet synchronous motors. However, this paper lacks research on the performance of Q-PLL under varying speed conditions. In [23], a two-phase phase loop structure called TP-PLL is proposed to solve the influence of harmonics in the back-EMF. The TP-PLL is used to integrate the EMF of the stator flux linkage. According to the measured current and the rotor flux, the equations are solved to obtain the angular position and amplitude of the stator flux. This algorithm can alleviate the DC offset problem on the pure integrator and suppress the influence of harmonic components on the position accuracy; however, the selection of its parameters lacks the support of frequency characteristics. In [24], by establishing a nonlinear equivalent model of SMO, a feed-forward PLL is introduced to analyze and compensate the position estimation deviation.

Aiming at the abovementioned problem of sliding mode control methods, this paper proposes a PLL based on a hybrid filter stage for PMSM rotor position estimation. Under the premise of satisfying the stability, a frequency-adaptive hybrid filtering stage-based PLL is proposed. The adaptive law of parameters in the proposed PLL is also presented. The proposed rotor position estimation method can remain stable even when the rotating speed changes. The parameter design and structure of the proposed PLL are demonstrated in detail. The experimental results show that the method has satisfactory disturbance rejection performance under harmonic interference. It can also predict the rotating speed and rotor position accurately, even under speed-changing conditions.

## 2. Rotor Position Estimation Based on PLL

The traditional PMSM sensorless control structure is shown in Figure 1. The estimated back-EMF is obtained by SMO. Then, it is fed to the proposed PLL with a hybrid filtering stage. The proposed hybrid filtering stage with adaptive parameters is responsible for rejecting harmonics in back-EMF. Hence, without the influence of harmonic EMF, the rotor position and rotating speed are estimated by the proposed PLL.

The voltage formula in *αβ*-frame can be expressed as
(1)$$\left[\begin{array}{c} u_{\alpha }\\ u_{\beta } \end{array}\right]=r_{s}\left[\begin{array}{c} i_{\alpha }\\ i_{\beta } \end{array}\right]+L_{s}p\left[\begin{array}{c} i_{\alpha }\\ i_{\beta } \end{array}\right]+\left[\begin{array}{c} e_{\alpha }\\ e_{\beta } \end{array}\right]$$
where *u_α_*、*u_β_*、*i_α_*、*i_β_*、*e_α_*、*e_β_* are stator voltage, stator current, and back-EMF in the *αβ*-coordinate system, respectively. *r_s_* and *L_s_* are the equivalent resistance and inductance in stator. The back-EMF term in Equation (1) contains rotor position information, which can be used to estimate the position of rotor. To extract rotating information of the rotor, the PLL is attached after SMO, as shown in Figure 1.

In Figure 2, *K_e_* is the gain of sliding mode, and sgn(·) is sign function. The core idea is to adjust the error between the estimated current value and the actual value. Its output is smoothed by a low-pass filter (LPF). The estimated values of the back-EMF are fed to the orthogonal PLL to obtain the estimated rotor position ${\hat{\theta }}_{r}$ and the rotating angular frequency ${\hat{\omega }}_{r}$. The characteristic of sgn(·) gives SMO strong robustness. However, it brings the problem of high-frequency harmonic pollution. Therefore, LPF must be introduced to cooperate with the orthogonal PLL.

Taking into account the harmonic components of stator voltages and currents, which may be caused by unbalanced rotor structure or unbalanced power supply, the oscillation errors are transferred into the PLL and appear in the estimation of rotor position and rotating frequency. By using the symmetrical component method, the components contained in the EMF are divided into positive sequence (PS) and negative sequence (NS). Applying Clark transformation, the EMF in *αβ*-frame is
(2)$$\left[\begin{array}{c} e_{\alpha }\\ e_{\beta }\\ e_{0} \end{array}\right]=\left[T_{\alpha \beta }\right]\left[\begin{array}{ccc} E_{1,4,7,\ldots }^{+}\sin\left(n\omega_{r}t+\theta_{r,n}\right) & +E_{1,2,5,8,\ldots }^{-}\sin\left(n\omega_{r}t+\theta_{r,n}\right) & E_{3,6,9,\ldots }^{0}\sin\left(n\omega_{r}t+\theta_{r,n}\right)\\ E_{1,4,7,\ldots }^{+}\sin\left(n\omega_{r}t+\theta_{r,n}-\frac{2\pi }{3}\right) & +E_{1,2,5,8\ldots }^{-}\sin\left(n\omega_{r}t+\theta_{r,n}+\frac{2\pi }{3}\right) & E_{3,6,9\ldots }^{0}\sin\left(n\omega_{r}t+\theta_{r,n}\right)\\ E_{1,4,7,\ldots }^{+}\sin\left(n\omega_{r}t+\theta_{r,n}+\frac{2\pi }{3}\right) & +E_{1,2,5,8,\ldots }^{-}\sin\left(n\omega_{r}t+\theta_{r,n}-\frac{2\pi }{3}\right) & E_{3,6,9,\ldots }^{0}\sin\left(n\omega_{r}t+\theta_{r,n}\right) \end{array}\right]$$

After applying Clark transformation *T_αβ_*, Equation (2) can be expressed as follows:(3)$$\left[\begin{array}{c} e_{\alpha }\\ e_{\beta } \end{array}\right]=\left[\begin{array}{c} E_{\alpha ,1,7,13,\ldots }^{+}(t)\\ E_{\beta ,1,7,13,\ldots }^{+}(t) \end{array}\right]+\left[\begin{array}{c} E_{\alpha ,1,5,11,\ldots }^{-}(t)\\ E_{\beta ,1,5,11,\ldots }^{-}(t) \end{array}\right]$$

By observing Equation (3), it is found that there is no third harmonic pollution in the *αβ*-coordinate system; only +1st, −1st, −5th, +7th, −11th, +13th… order sequence components exist. After Park transformation, the harmonics in *αβ*-coordinate system are converted into the even harmonics in *dq*-coordinate system, as shown in Table 1. The sign in the table indicates the rotating direction of EMF. Positive is the clockwise direction and vice versa.

## 3. The Proposed PLL

To achieve a satisfactory performance for rotor estimation, adaptive notch filters are employed in quasi-type-1 PLL (QT1-PLL) for better performance in the proposed method. QT1-PLL is derived from stationary reference frame PLL (SRF-PLL) and moving average filter-based PLL (MAF-PLL). These three PLLs are depicted in Figure 3.

${\hat{e}}_{\alpha \beta }$ represents estimated EMF. *e_d_* and *e_q_* are the components in *dq*-frame. *ω_r_* represents the nominal angular rotating speed. *Δω* is the deviation of rotating speed. $\hat{\omega }$ and ${\hat{\theta }}_{r}$ are the estimated value of rotating speed and rotor position. According to [25], the open loop transfer functions can be written as
(4)$$G_{olSRF}(s)=(k_{p}+\frac{k_{i}}{s})\frac{1}{s}$$
(5)$$MAF(s)=\frac{1-e^{-T_{\omega }s}}{T_{\omega }s}\approx \frac{1}{\frac{T_{\omega }}{2}s+1}$$
(6)$$G_{olMAF}(s)=MAF(s)(k_{p}+\frac{k_{i}}{s})\frac{1}{s}\approx \frac{1}{\frac{T_{\omega }}{2}s+1}(k_{p}+\frac{k_{i}}{s})\frac{1}{s}$$
(7)$$G_{olQT1}(s)=(\frac{MAF(s)}{1-MAF(s)})(k_{p}+\frac{k_{i}}{s})\approx \frac{2}{T_{\omega }s}(k_{p}+\frac{k_{i}}{s})$$

*k_p_* and *k_i_* are the parameters in PI controller. The Bode diagram of three PLLs under fixed rotating speed is depicted in Figure 4. From the approximation in Equations (6) and (7), it is noticed that the pole points in the QT1-PLL and SRF-PLL are less than that in the MAF-PLL, which means the bandwidth of these two PLLs are smaller than that in the MAF-PLL. At the same time, the QT1-PLL can also provide a satisfactory filtering capability, as the MAF-PLL does. It can totally remove the dominant harmonic components. However, the time delay existing in MAF, which lasts for 0.01 s, reduces the response speed during transient behavior.

Motivated by the advantage of the QT1-PLL, a hybrid filtering stage-based PLL is proposed in this section. The schematic of the proposed method is illustrated in Figure 5. Adaptive notch filters (ANFs) and MAFs are utilized as a hybrid filtering stage which is arranged after Park transformation for harmonic elimination at *dq*-frame. ANFs are adopted to remove the −2*ω_r_* component. MAFs are used for removing the rest of the harmonics. Without the task of removing −2*ω_r_*, the window length (*T_ω_*) is reduced to π/(3${\hat{\omega }}_{r}$).

The mathematical equation of ANF after Laplace transformation is
(8)$$ANF(s)=\frac{s^{2}+{\left(2{\hat{\omega }}_{r}\right)}^{2}}{s^{2}+2{\hat{\omega }}_{r}\xi s+{\left(2{\hat{\omega }}_{r}\right)}^{2}}$$

The damping factor *ξ* is 0.7. According to Equation (5), the hybrid filtering stage is
(9)$$H(s)=ANF(s)MAF(s)=\frac{s^{2}+{\left(2{\hat{\omega }}_{r}\right)}^{2}}{s^{2}+2{\hat{\omega }}_{r}\xi s+{\left(2{\hat{\omega }}_{r}\right)}^{2}}\frac{1-e^{-T_{w}s}}{T_{w}s}$$

Hence, the frequency characteristic can be calculated by Equation (9). The corresponding Bode plot of Equation (9) is shown in Figure 6. Since the proposed method is working under varying speed conditions, the filtering stage is designed to be frequency-adaptive with rotating speed of the rotor. The frequency-adaptive MAF is implemented by setting window length (*T_ω_*) to be π/(3${\hat{\omega }}_{r}$), which is adaptive with ${\hat{\omega }}_{r}$. The adaptive ANF is achieved by using a frequency-adaptive structure depicted in Figure 7.

According to the block diagram of the proposed PLL, the small-signal model can be obtained, as shown in Figure 8. By applying some block diagram algebraic, the small-signal model can be changed to the schematic, as shown in Figure 9, which is a typical closed-loop system. Then, the open-loop transfer function of the proposed method can be written as
(10)$$G_{ol}(s)=\frac{{\hat{\theta }}_{r,1}^{+}(s)}{\theta_{r,1}^{+}(s)-{\hat{\theta }}_{r,1}^{+}(s)}=(\frac{H(s)}{1-H(s)})(\frac{s+k}{s})$$

The transfer function of position estimation error can be expressed as
(11)$$G_{e}(s)=\frac{\theta_{e}(s)}{\theta_{r,1}^{+}(s)}=\frac{\theta_{r,1}^{+}(s)-{\hat{\theta }}_{r,1}^{+}(s)}{\theta_{r,1}^{+}(s)}=\frac{1}{1+G_{ol}(s)}$$
where position estimated error is *θ_e_*. Hence, the response of *θ_e_* when a step change suddenly occurs in rotor position can be expressed as
(12)$$\Theta_{e}^{\Delta \theta }(s)=\frac{\Delta \theta }{s}G_{e}(s)$$

The response of *θ_e_* when a step change suddenly occurs in rotating speed can be expressed as
(13)$$\Theta_{e}^{\Delta \omega }(s)=\frac{\Delta \omega }{s^{2}}G_{e}(s)$$

To achieve a satisfactory dynamic performance when rotor position and rotating speed suddenly change, inverse Laplace transformation are applied to both Equations (12) and (13). Then, the variations of 2% settling time as a function of *k* under different rotating speeds (500, 1000, and 2800 rpm) can be depicted in Figure 10, Figure 11 and Figure 12. From the observation of Figure 10, Figure 11 and Figure 12, it is impossible to obtain the best performance under both conditions. A trade-off of *k* should be made for a good performance under both conditions. When the rotating speed is 500 rpm, the best choices of *k* for both conditions are *k* = 48 and *k* = 54, respectively. Hence, *k* should be selected between 48 and 54. In this paper, *k* is set to be 50. In addition, when the rotating speed changes, the value of *k* should follow. When the rotating speed is 1000 rpm, similarly, *k* is determined as 130, according to Figure 11. When the speed climbs to 2800 rpm, *k* is selected to be 290.

To achieve a good performance under both conditions, an optimal value of *k* is better to be chosen around the bottom of both curves in these figures. It can be observed that the optimal *k* under different rotating speeds is different. The relationship between rotating speed and optimal value of *k* is depicted in Figure 13. According to this relationship, the parameter *k* in the proposed PLL is adaptive with the rotating speed from 20 to 290 under 300 rpm to 2800 rpm.

To verify the stability of the proposed PLL, Nyquist plots are examined with different *k* corresponding to different rotating speeds, as shown in Figure 14. All curves do not encircle the (−1, j0) point, which means the system is stable with different *k*.

The noise pollution is an important factor which may influence the estimation performance. To examine the capability of noise immunity of the proposed method, a simulation is carried out when the currents of stators *αβ*-frame are polluted by white noise. The simulation results are shown in Figure 15. It can be seen that there are no oscillation errors in estimated rotating speed. Owing to the filtering capability for high-frequency noise, the proposed method is immune to white noise which may occur in the procedure of sampling behavior.

Moreover, besides its use in rotor position estimation, the proposed PLL can also be used in other power converter-based devices for grid-connected applications. According to Figure 5, it can be seen that the proposed PLL can estimate the frequency and phase of input signal as long as the input signal is a sinusoidal periodical vector. It only needs to feed the vector of grid voltages at *αβ*-frame to extract the phase and frequency of the power system.

## 4. Experimental Results

In order to prove the effectiveness of the proposed method, experimental studies were carried out on a 550 W sensorless PMSM control system. The number of pole pairs of the PMSM used in the experiment was 2. The resistance of the stator was 0.63 Ω. The inductance of the direct axis and the orthogonal axis were 119 mH and 29 mH, respectively. The encoder was installed at the end of the load. The PMSM was driven by an inverter-based motor driver controlled by a DSP. The switching frequency of inverter was 10 kHz. The sensors sampled data at the beginning of each switching period. The waveform of estimated position and EMFs were monitored in the Labview software environment on a host computer. The parameters are summarized in Table 2. The experimental setup is shown in Figure 16.

The proposed PLL was implemented in discrete domain on a DSP board, while it was designed in continuous domain. A discrete time realization is required for practical applications. By using backward Euler method, the discrete approximation of integral operation is as follows:(14)$$\frac{1}{s}\Leftrightarrow \frac{T_{S}}{1-z^{-1}}$$

The discretization effect of the proposed PLL was examined by a simulation. The sampling frequency used in the discretization was 10 kHz. Figure 17 illustrates the discretization effect when phase jump and frequency step change occur to back-EMF at the input of the PLL. As shown in Figure 17, the transient behaviors in the continuous domain and discrete domain are almost the same.

To simulate the actual situation of the motor and speed change, the initial speed of the motor was set to 800 rpm. The experimental study included the proposed PLL, QT1-PLL, and MAF-PLL. To be fair, the MAF was used in the QT1-PLL, and the MAF-PLL is adaptive with rotating period. According to the design guidelines in [25], the only parameter *k* in the QT1-PLL was fixed, at 92. The parameters in the MAF-PLL were designed according to the guidelines in [26]. Both the MAF-PLL and QT1-PLL were not only used in sensorless control, but also used in grid synchronization for power converters.

The estimation performance was studied under speed changing conditions. The rotating speed of PMSM was 800 rpm at the first stage. After that, the rotating speed of PMSM was reduced to 50 rpm to examine the performance under low-speed condition. Then, the rotating direction was reversed to test the performance under such a condition. The curve of rotating speed measured by the encoder is depicted in Figure 18.

Figure 19 illustrates the stator current *i_α_*. The waveform contains amounts of harmonics and noise. The presence of harmonics may pollute the waveform and degrade the performance of estimation.

Figure 20 shows the rotor position by using the encoder. Figure 21 shows the comparison of the proposed PLL, QT1-PLL, and MAF-PLL when the PMSM is at deceleration process. It can be seen that the estimation of the proposed PLL is closer to the actual value than other PLLs’. It confirms that the transient response of the proposed PLL is faster. The QT1-PLL can also provide a good transient speed. However, the window length of the MAFs used in the QT1-PLL makes its converge time larger than the proposed PLL. The response time of the MAF-PLL is very slow, which results in a large estimation error in transient behavior.

Figure 22 illustrates the comparison between estimated speed of the proposed PLL and actual rotating speed. It can be seen that during almost every stage of the case study, the proposed PLL can accurately estimate the rotating speed of the PMSM. Figure 23 depicts the estimated error during the entire experimental study. The peak errors happen in the acceleration and deceleration procedure, which can hardly exceed 10 rpm. The errors during low-speed conditions are less than 1. The key reason for such a satisfactory performance is the well-designed adaptive control parameter. The control parameter *k* and ANF with frequency-adaptive structure make the proposed PLL adaptive with rotating speed, which can provide optimal parameters for the entire operating conditions.

Figure 24 illustrates the comparison between estimated speed of the MAF-PLL and actual rotating speed. It can be seen that the estimated error around acceleration and deceleration procedure are larger than that in the proposed PLL. Figure 25 depicts the estimated error during the entire experimental study. The peak errors happen in the acceleration and deceleration procedure, which are much bigger than those in the proposed PLL, which are larger than 40 rpm. Although MAFs used in the MAF-PLL are also adaptive, the parameters in its PI controller are not adaptive with rotating speed. Large overshoots occur when reference speed changes. A PI controller with fixed parameters is not suitable for such conditions. Even so, the MAF-PLL can also attenuate harmonic and noise effectively.

Figure 26 illustrates the comparison between estimated speed of the QT1-PLL and actual rotating speed. It can be seen that the QT1-PLL can track actual rotating speed rapidly without any overshoot. However, oscillation errors in estimation exist during almost every stage of the case. Figure 27 depicts estimated error of the QT1-PLL. It can be seen that oscillation errors are much larger than those in the other PLLs, which are more than 15 rpm. Although the transient speed of the QT1-PLL is as good as the proposed PLL, without using adaptive control parameter, the disturbance attenuation capability is degraded under changing speed conditions.

In sum, owing to the usage of frequency-adaptive parameter changing strategy, the proposed PLL can provide an accurate estimation performance and disturbance rejection capability. Although the harmonic filtering capability of the MAF-PLL is also satisfactory, the overshoots existing in acceleration and deceleration illustrate that its dynamic performance is deteriorated when speed changes. The disturbance rejection capability is highly degraded when the control parameters are fixed during changing speed conditions. All the mentioned points above reveal that the proposed PLL with frequency-adaptive parameters is suitable for rotor position estimation.

## 5. Conclusions

This paper proposes a PLL based on a hybrid filter stage for PMSM rotor position estimation. Frequency-adaptive notch filters and MAFs are adopted in the inner loop of the PLL, which can totally eliminate harmonic sequences. To ensure the stability and achieve satisfied dynamic performance under varying rotating speeds, the control parameter of the proposed PLL is set to be adaptive with rotating speed. To validate the effectiveness of the proposed method, comparative simulations were carried out. The results show that the proposed method can reduce the impact from harmonic EMF. The proposed method can provide a satisfactory position estimation performance under wide speed changing conditions without deterioration by harmonic EMF.

## Figures and Tables

**Figure 1 sensors-21-04667-f001:**
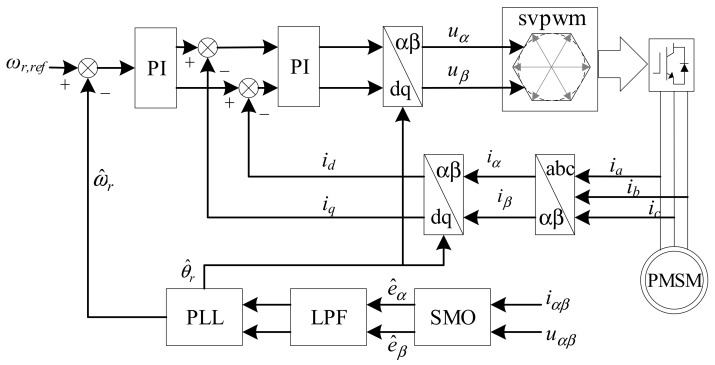
The structure of PMSM sensorless control.

**Figure 2 sensors-21-04667-f002:**
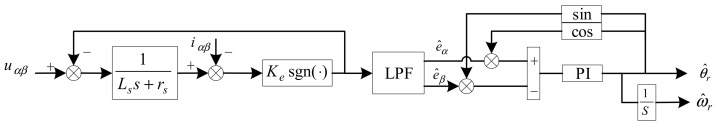
The schematic of back-EMF observer based on SMO.

**Figure 3 sensors-21-04667-f003:**
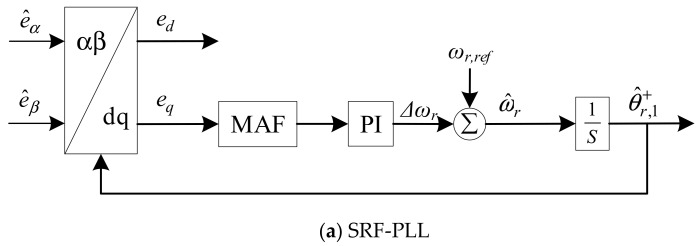
The structure of three PLLs.

**Figure 4 sensors-21-04667-f004:**
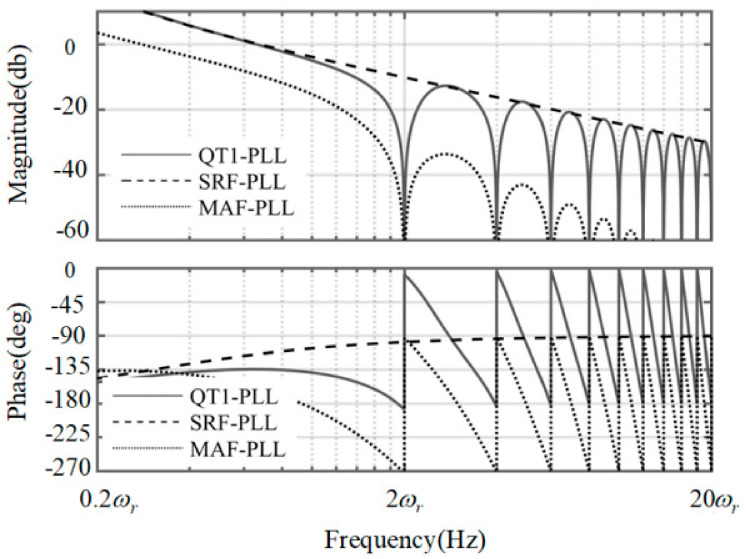
The Bode plot of three PLLs under fixed rotating speed.

**Figure 5 sensors-21-04667-f005:**
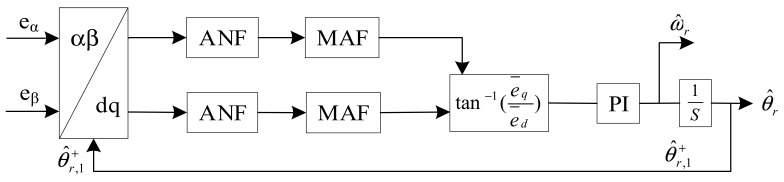
Block diagram of the proposed PLL.

**Figure 6 sensors-21-04667-f006:**
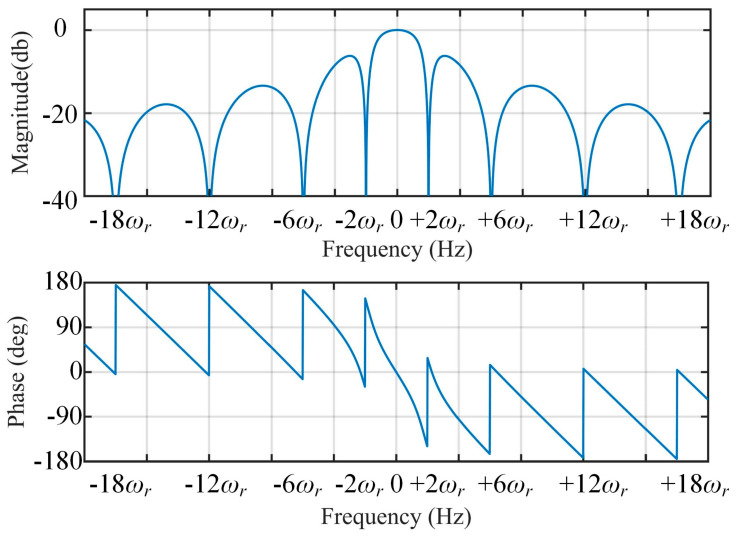
Bode diagram of *H*(*s*).

**Figure 7 sensors-21-04667-f007:**
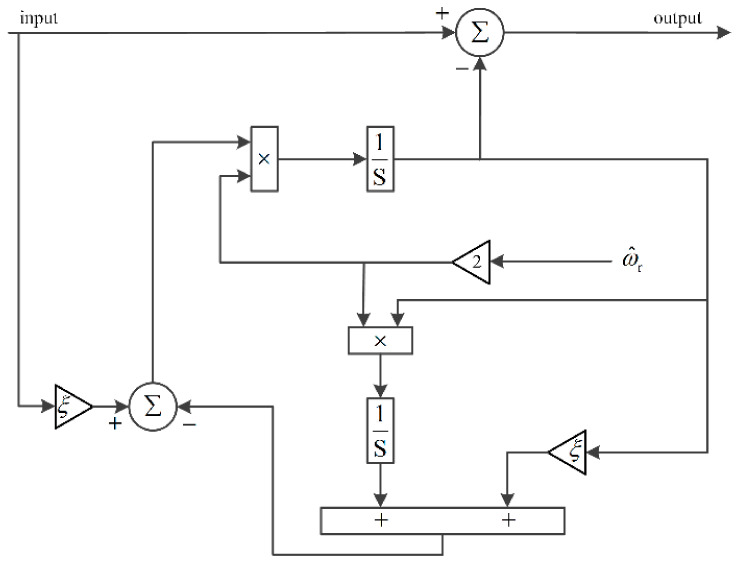
The frequency-adaptive implementation of ANF.

**Figure 8 sensors-21-04667-f008:**
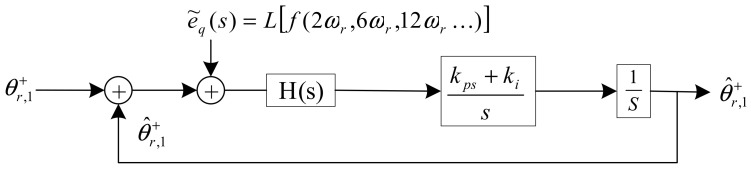
Small-signal model of the proposed PLL.

**Figure 9 sensors-21-04667-f009:**
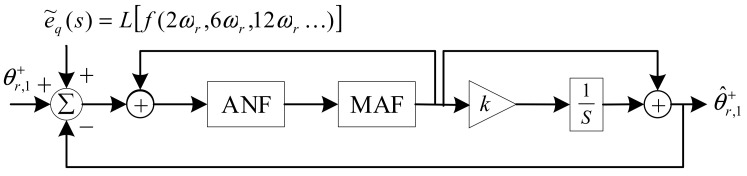
The simplified model of the proposed PLL.

**Figure 10 sensors-21-04667-f010:**
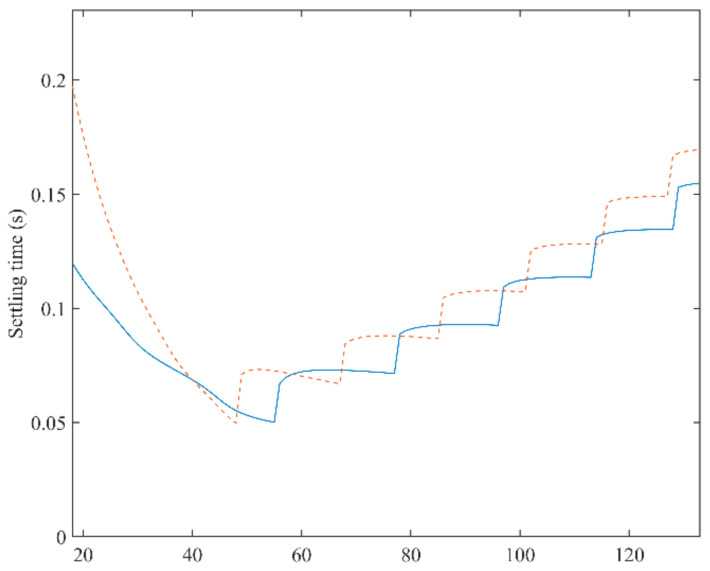
A 2% settling time of the proposed PLL as a function of *k* for both phase jump (solid line) and frequency step change (dashed line) when rotating speed is 500 rpm.

**Figure 11 sensors-21-04667-f011:**
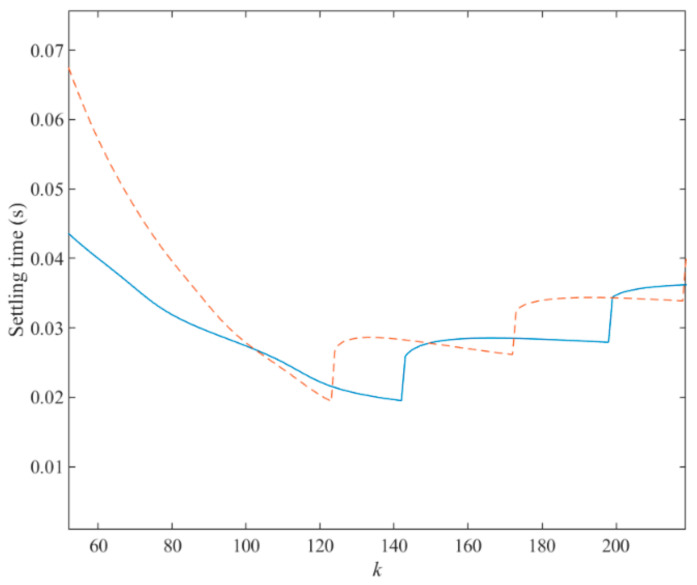
A 2% settling time of the proposed PLL as a function of *k* for both phase jump (solid line) and frequency step change (dashed line) when rotating speed is 1000 rpm.

**Figure 12 sensors-21-04667-f012:**
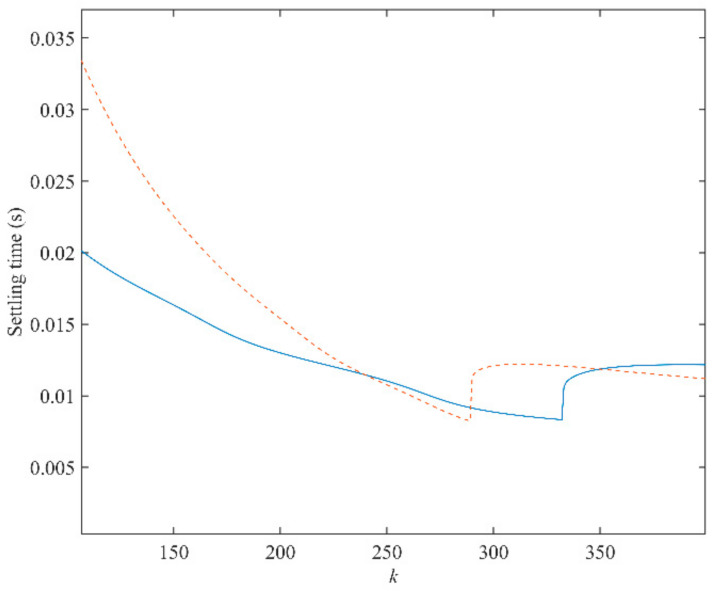
A 2% settling time of the proposed PLL as a function of *k* for both phase jump (solid line) and frequency step change (dashed line) when rotating speed is 2800 rpm.

**Figure 13 sensors-21-04667-f013:**
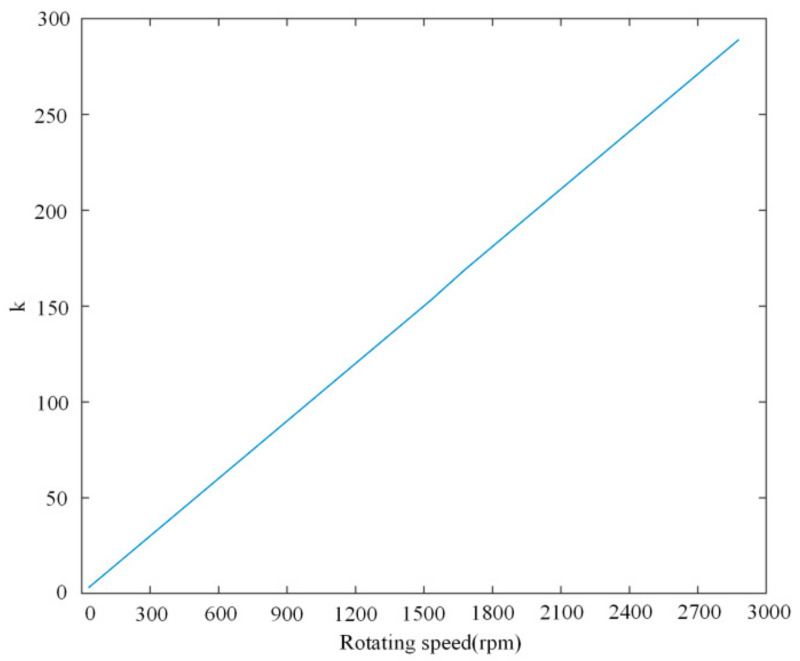
Optimal value of *k* as a function of rotating speed.

**Figure 14 sensors-21-04667-f014:**
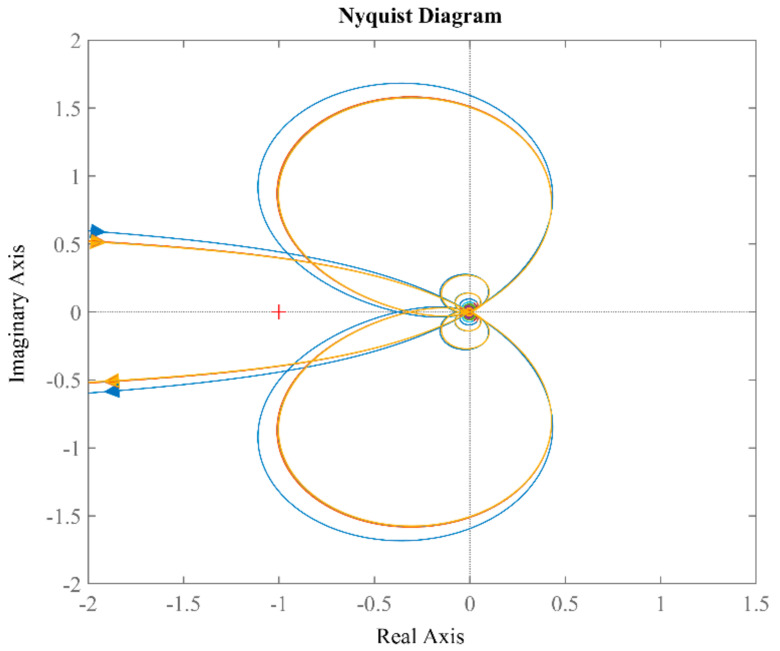
Nyquist curves of *G_ol_*(*s*) with *k* = 50, 130, 290.

**Figure 15 sensors-21-04667-f015:**
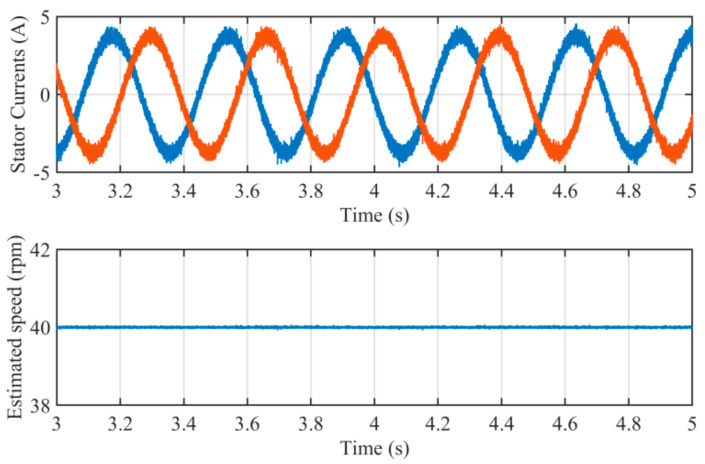
Simulation results when the stator currents are polluted by white noise.

**Figure 16 sensors-21-04667-f016:**
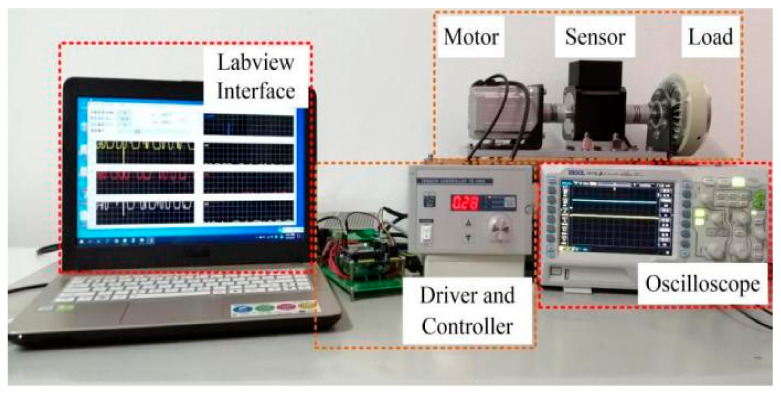
Experimental setup of control system.

**Figure 17 sensors-21-04667-f017:**
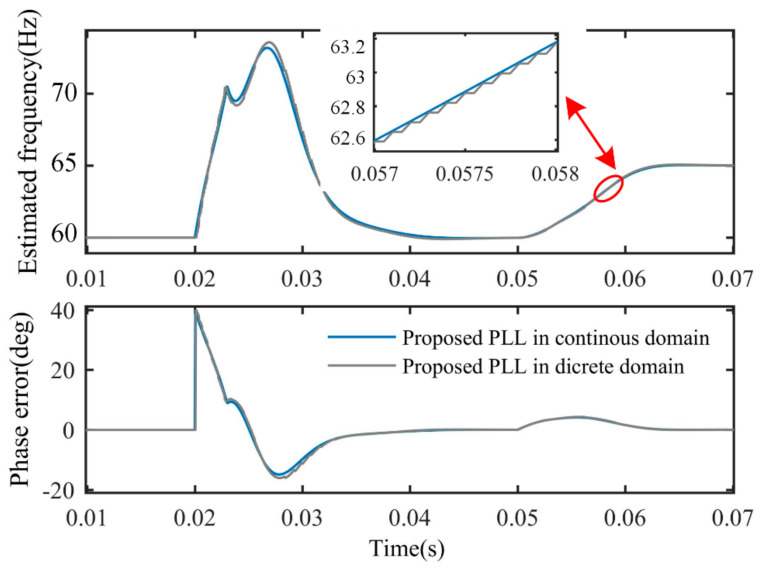
Discretization effect of the proposed PLL when disturbances occur to back-EMF.

**Figure 18 sensors-21-04667-f018:**
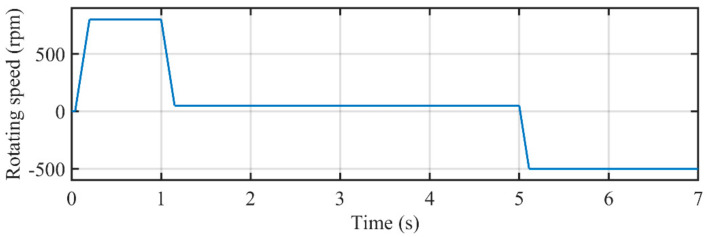
The rotating speed measured by the encoder.

**Figure 19 sensors-21-04667-f019:**
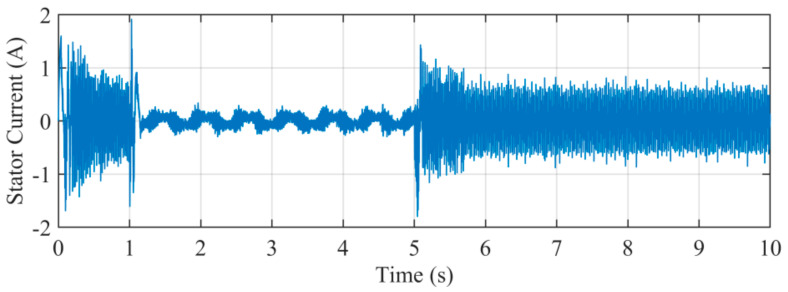
Stator current *i_α_* under experiment.

**Figure 20 sensors-21-04667-f020:**
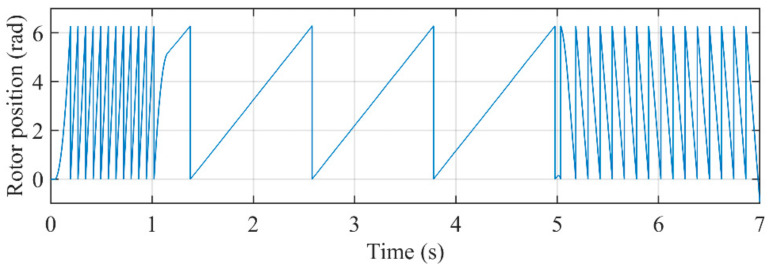
Actual rotor position.

**Figure 21 sensors-21-04667-f021:**
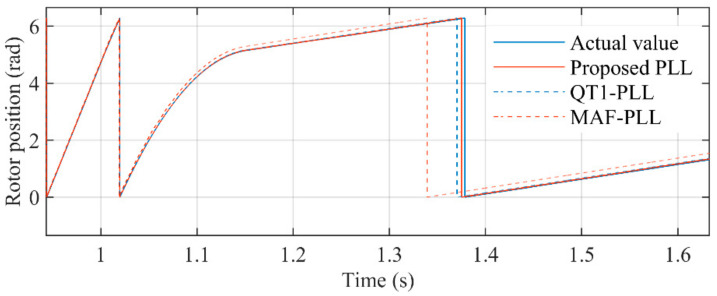
Estimated position when the speed is being reduced to low speed.

**Figure 22 sensors-21-04667-f022:**
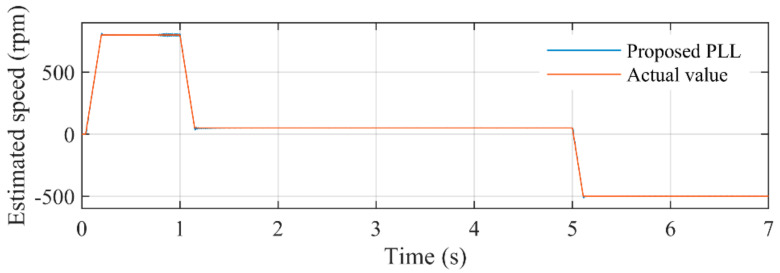
The comparison between estimated rotating speed by the proposed PLL and the actual speed.

**Figure 23 sensors-21-04667-f023:**
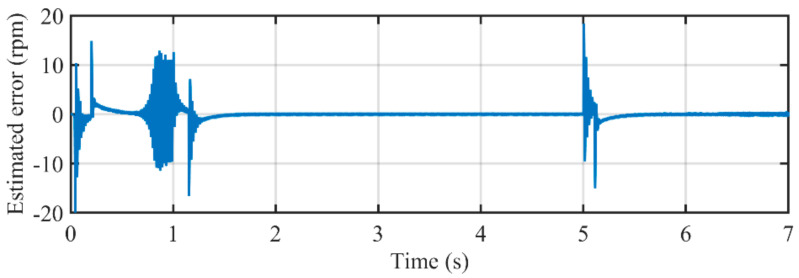
Estimated error in rotating speed by using the proposed PLL.

**Figure 24 sensors-21-04667-f024:**
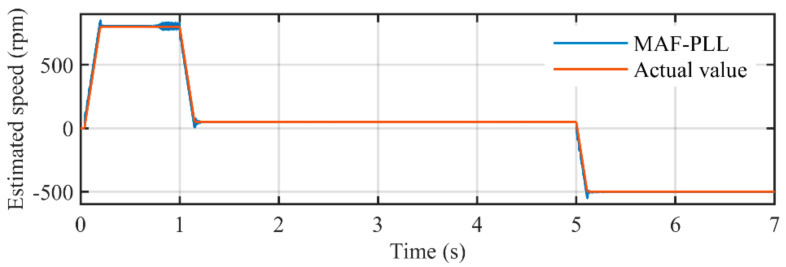
The comparison between estimated rotating speed by the MAF-PLL and the actual speed.

**Figure 25 sensors-21-04667-f025:**
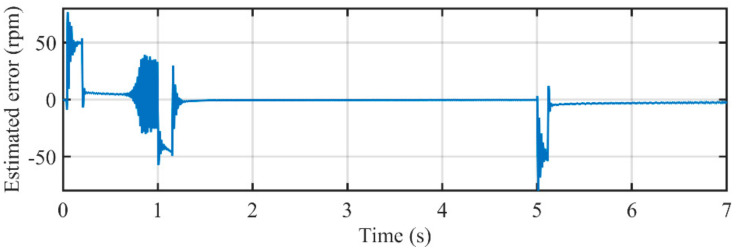
Estimated error in rotating speed by using the MAF-PLL.

**Figure 26 sensors-21-04667-f026:**
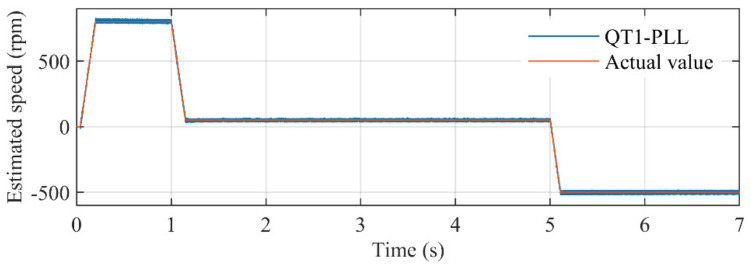
The comparison between estimated rotating speed by the QT1-PLL and the actual speed.

**Figure 27 sensors-21-04667-f027:**
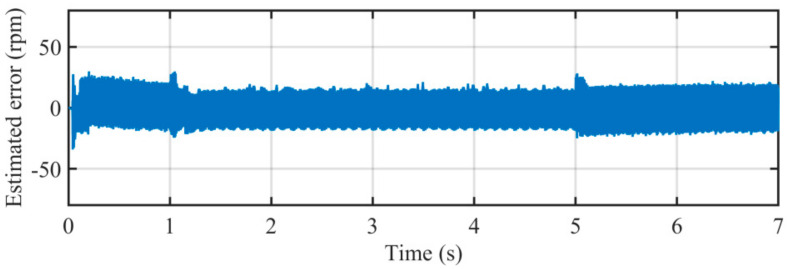
Estimated error in rotating speed by using the QT1-PLL.

**Table 1 sensors-21-04667-t001:** EMF sequence components.

Harmonic Order	*αβ* Frame (Rad/s)	Harmonic Order	*dq* Frame (Rad/s)
…	…	…	…
+1	*ω_r_*	0	0
−1	−*ω_r_*	−2	−2 *ω_r_*
−5	−5 *ω_r_*	−6	−6 *ω_r_*
+7	7 *ω_r_*	+6	6 *ω_r_*
−11	−11 *ω_r_*	−12	−12 *ω_r_*
+13	13 *ω_r_*	12	12 *ω_r_*
…	…	…	…

**Table 2 sensors-21-04667-t002:** The parameters of the PMSM.

Rated power	550 W	Stator resistance	0.63 Ω
Rated current	2.6 A	d-axis inductance	119 mH
Rated speed	4000 rpm	q-axis inductance	29 mH
Rated torque	5 N·m	Pole pairs	2
Rated frequency	50 Hz	Inertia	0.011 kg·m^2^

## Data Availability

Not applicable.

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
