# Peer review of "A Hybrid Filtering Stage-Based Rotor Position Estimation Method of PMSM with Adaptive Parameter"

_sensors, 2021, doi:10.3390/s21144667_

Round 1
Reviewer 1 Report
The paper is well organized and very interesting. Presented method of controlling the rotor position is valuable. Experiment proved the theoretical assumptions, and value of this action is very high.
The Introduction chapter is well prepared and fullfill the necessary literature review. Hovewer I have some comments.
There are some editorial mistakes. Please check necessary dots, spaces and commas.
In conclusions the gains of using the proposed contriol system should be detailed and highlighted.
Author Response
Dear reviewer
Thanks for your valuable comments. Please check the attachment.
Best Regard
All authors

Reviewer 2 Report
- Proof reading needed: Due to the advantages of efficiency and high power density, permanent magnet synchronous motors (PMSM) has->have been widely used in electric vehicles, petroleum and other industrial fields [1-3] .
- The measurement setup is poorly described.
- The measurement uncertainty of the measurement in not described nor discussed.
Reviewer 3 Report
This paper proposes a hybrid filtering stage based PLL to improve the rotor position estimation in a PMSM. Unfortunately, in this reviewer’s opinion, it lacks the following aspects:
-Proposed hybrid PLL is described in frequency domain, but implementation is discrete. How digital limitations affect the proposed one?
-Reported comparison is only between the proposed PLL and traditional SRF-PLL, which in this reviewer’s opinion is not fair. Add at least one or two more popular PLL to the comparison section. Also, add details of PLL’s tuning.
-Details of the test bed are missed.
-References are poorly written.
-Please add a discussion oh how the proposed hybrid PLL can be applied to other power electronics’ PLL application.
-The major problem of sensorless control is low speed and speed change direction. Why no tests under those scenarios are reported on the proposed strategy? Those tests are mandatory.
Reviewer 4 Report
In this paper is proposed a phase-locked loop (PLL) based on a hybrid filter stage for permanent magnet synchronous machine (PMSM) rotor position estimation. Adaptive notch filters and moving average filters are integrated together to eliminate harmonic electromotive forces. To ensure stability under varying rotating speed, the control parameter is set to be adaptive with rotating speed. Some comparative simulations are carried out.
My comments are as follows:
- The approximation (5) is true only if Tωs is small. But from the bode diagram (Fig.6) it can be seen that the approximations (6) and (7) are questionable.
- The error of estimated position (fig.22) for Time(s) [0.7, 0.9] is not small and large deviations are observed between proposed method and conventional method. Some comments and explanations are needed to clarify the results. It should be explained more clear how the proposed PLL can provide a better harmonic suppression capability.
- English should be polished throughout the paper to make it completely understandable. For example, at row 11 instead of “permanent magnetic synchronous machine” should be „permanent magnet synchronous machine”.
Round 2
Reviewer 3 Report
Thanks to reply my recommendations, the manuscript was improved. After a final english editing, it can be accepted.
Author Response
Thanks for review's comments. We revise some english writing mistakes this round.
Please see the attachment.
